# Post Natal Microbial and Metabolite Transmission: The Path from Mother to Infant

**DOI:** 10.3390/nu16131990

**Published:** 2024-06-22

**Authors:** Juan Manuel Vélez-Ixta, Carmen Josefina Juárez-Castelán, Daniela Ramírez-Sánchez, Noemí del Socorro Lázaro-Pérez, José Javier Castro-Arellano, Silvia Romero-Maldonado, Enrique Rico-Arzate, Carlos Hoyo-Vadillo, Marisol Salgado-Mancilla, Carlos Yamel Gómez-Cruz, Aparna Krishnakumar, Alberto Piña-Escobedo, Tizziani Benitez-Guerrero, María Luisa Pizano-Zárate, Yair Cruz-Narváez, Jaime García-Mena

**Affiliations:** 1Departamento de Genética y Biología Molecular, Cinvestav, Av. Instituto Politécnico Nacional 2508, Mexico City 07360, Mexico; juan.velez@cinvestav.mx (J.M.V.-I.); carmen.juarez@cinvestav.mx (C.J.J.-C.); danielaramireza23@gmail.com (D.R.-S.); noemi.lazaro@cinvestav.mx (N.d.S.L.-P.); aparna.krishnakumar@cinvestav.mx (A.K.); apinae@cinvestav.mx (A.P.-E.); tizziani.benitez@cinvestav.mx (T.B.-G.); 2Laboratorio de Posgrado e Investigación de Operaciones Unitarias, Escuela Superior de Ingeniería Química e Industrias Extractivas, Instituto Politécnico Nacional, Mexico City 07738, Mexico; jjcastro@ipn.mx (J.J.C.-A.); ericoarz@yahoo.com (E.R.-A.); marisol.salgado.mancilla98@gmail.com (M.S.-M.); cyamelgmz@outlook.com (C.Y.G.-C.); 3Unidad de Cuidados Intermedios al Recién Nacido, Instituto Nacional de Perinatología, Secretaría de Salud, Mexico City 11000, Mexico; silviarmzeta@yahoo.com.mx; 4Departamento de Farmacología, Cinvestav, Av. Instituto Politécnico Nacional 2508, Mexico City 07360, Mexico; citocromo@cinvestav.mx; 5Coordinación de Nutrición y Bioprogramación, Instituto Nacional de Perinatología, Secretaría de Salud, Mexico City 11000, Mexico; 6Unidad de Medicina Familiar No. 4, Instituto Mexicano del Seguro Social, Mexico City 06720, Mexico

**Keywords:** metabolome, microbiome, microbiota, 16S RNA gene, mass spectrometry, entero–mammary pathway, QIIME, FT-ICR MS

## Abstract

The entero–mammary pathway is a specialized route that selectively translocates bacteria to the newborn’s gut, playing a crucial role in neonatal development. Previous studies report shared bacterial and archaeal taxa between human milk and neonatal intestine. However, the functional implications for neonatal development are not fully understood due to limited evidence. This study aimed to identify and characterize the microbiota and metabolome of human milk, mother, and infant stool samples using high-throughput DNA sequencing and FT-ICR MS methodology at delivery and 4 months post-partum. Twenty-one mothers and twenty-five infants were included in this study. Our results on bacterial composition suggest vertical transmission of bacteria through breastfeeding, with major changes occurring during the first 4 months of life. Metabolite chemical characterization sheds light on the growing complexity of the metabolites. Further data integration and network analysis disclosed the interactions between different bacteria and metabolites in the biological system as well as possible unknown pathways. Our findings suggest a shared bacteriome in breastfed mother–neonate pairs, influenced by maternal lifestyle and delivery conditions, serving as probiotic agents in infants for their healthy development. Also, the presence of food biomarkers in infants suggests their origin from breast milk, implying selective vertical transmission of these features.

## 1. Introduction

In recent years, the vertical mother–newborn transmission of maternal gut bacteria by breastfeeding has been studied [1,2,3,4], and more recently the entero–mammary pathway (EMP) theory also known as the Gut–Breast Axis has been considered in detail [5,6,7,8,9,10,11]. The molecular mechanism of the EMP in humans has been demonstrated by the capacity of the dendritic cells, to pull viable beneficial bacteria from the maternal intestinal lumen. These bacteria are transferred to the mammary gland and eventually into the milk for microbiome establishment in the neonate [3,12,13]. From an evolutionary perspective, the EMP and the composition of human milk (HM) are important due to certain bacterial species that undergo strong selection to initiate the colonization of the intestinal microbiota of the newborn for subsequent maturation [14].

Even though it is well known that breastfeeding has a key role in the optimal health and development of the immune system of the newborn [15,16], a low breastfeeding rate is not uncommon in both low- and middle-income countries, with only 37% of newborns younger than 6 months exclusively breastfed [17]. Thus, it is important to emphasize that exclusive HM feeding for the first 6 months of life, with continued breastfeeding for 1 to 2 years of life or longer, is recognized and recommended as the normative standard for infant feeding [18,19].

HM has benefits to the neonate like improved neurodevelopmental outcomes, decreased infection risk during neonatal and infant periods, and decreased childhood obesity [20]. HM contains multiple immunologically important compounds, maternal immunoglobulins, maternal immune cells, and antimicrobial compounds including free fatty acids, lactoferrin, and lysozyme. It also contains nutritional components including casein, lactose, fat, vitamins, and minerals, and different bioactive components such as functional proteins, miRNAs, immunoregulatory factors, immune cells, and specific microbiota bacteria [21,22].

Additional important compounds of the HM are diverse metabolites that can be transferred through the EMP to the newborn, being a relevant element in the infant’s health. However, the study of the HM metabolome is scarce, with only a few reports [23,24,25]. These studies suggest that the HM metabolome undergoes significant changes during lactation and may play a critical role in infant growth and development. In addition, short-chain fatty acids (SCFAs) are the main end products of the metabolism of bacteria attached to surfaces of the intestinal lumen [26], and they are associated with metabolic parameters in newborns and mothers [27]. SCFAs were identified as well as a potential biomarker for early screening of pre-eclampsia [28]. Through pregnancy, SCFAs are products of bacterial metabolism, responsible for preserving homeostasis in the woman’s body, as well as influencing immune function and carbohydrate and lipid metabolism. It is also important to mention that metabolic changes occurring during pregnancy contribute considerably to changes in the phylogenetic diversity of the gut microbiota, which differs in several stages of pregnancy [29].

For the microbiota diversity in HM, several of these studies indicate that most of these bacterial communities are anaerobic. Additionally, it has been established that the diversity varies during the lactation stages, and also the HM composition varies among individuals, depending on many factors, such as maternal diet, genetics, health, antibiotic usage, demography, environmental differences and mode of delivery [30,31,32]. Differences in microbiota composition depending on birth mode affect the infant microbial colonization, which is altered in infants delivered by C-section compared to vaginally born infants impacting their health [33]. Nine genera differentiated the milk samples of women who gave birth vaginally or delivered by elective C-section. Vaginal deliveries were characterized by *Peptoniphilis*, *Anaerococcus*, *Listeria*, and an unclassified bacterium, whereas C-section deliveries were associated with *Enterococcus*, *Enterobacter*, *Rhodoccus* and *Burkholderia* [32]. The microbial delivery in HM is different in women with elective C-section [33]. Nonetheless, it is reported that *Staphylococcus*, *Streptococcus*, *Serratia*, *Pseudomonas*, *Corynebacterium*, *Ralstonia*, *Propionibacterium*, *Sphingomonas*, and *Bradyrhizobium* are nine genera that constitute the “core” bacteriome of human milk microbiota in this study. Remarkably, these nine ‘‘core’’ OTUs represented about half of the microbial community observed. They also concluded that human milk contains a collection of bacteria more diverse than previously reported [22]. Another report showed that emergency C-section milk samples harbor a similar microbial profile to that in vaginal deliveries while non-emergency showed a microbial profile similar to the skin and oral microbial communities [34]. Cesarean-born infants presented lower abundances of *Bacteroides*, lower prevalence of *Bifidobacterium longum*, and higher abundances of *Akkermansia* and *Kluyvera*. The overall microbiota diversity (alpha and beta) was not different between cesarean and vaginally born infants fed with HM [35].

Although the presence of microbiota and metabolites in HM is well-documented, there is a lack of studies that thoroughly characterize their composition and similarities across the mother’s colon, HM, and the newborn’s colon. Additionally, the extent of the microbiota’s impact on the metabolomic profile of both the mother’s and the infant’s gut, along with the degree of transmission of these substances, remains poorly understood. Studying these components in mothers and infants would offer a better insight into the importance and impact of human milk on the healthy maturation and development of the newborn. This study presents a time-based characterization not only of the microbiota but also of the metabolome from HM and stool samples from a cohort of healthy pregnant women and their neonates.

## 2. Materials and Methods

### 2.1. Study Design and Selection of Subjects

An observational and longitudinal study was conducted with Mexican pregnant women and infants receiving medical care at the “Instituto Nacional de Perinatología-Isidro Espinosa de los Reyes”, a third-level governmental hospital located in Mexico City (19°25′22″ N 99°12′21″ W). Samples were collected between June 2022 and August 2023. Inclusion criteria for participants were: Mexican origin for at least two generations, women in the last trimester of pregnancy, between 16 and 45 years old, spontaneous vaginal delivery or non-elective C-section, singleton pregnancy, full-term pregnancy, offspring without congenital malformations. The exclusion criteria were probiotics and alcohol consumption, smoking, diabetes, overweight and obesity before or during pregnancy, antibiotics use during the last trimester of pregnancy, regular consumption of medications such as anti-inflammatories, laxatives, antacids prior to sample collection, hormone treatment, gastrointestinal disease or other pathologies. This study was approved by the hospital’s Research, Biosafety and Ethics Committee with the register number INPER-DG-000419-2019_2018-1-172. All participants consented to the collection of data and signed informed consent following the Declaration of Helsinki.

### 2.2. Data and Specimen Collection

Participants were healthy lactating women, and full-term exclusively breastfed infants. Milk and fecal samples were collected from mothers and infants between 1- and 2-days *post-partum*, and 4 months after birth. A total of 25 fecal samples from mothers (MS), 32 fecal samples from infants (IS), and 26 HM samples were collected. HM for this study was manually collected into a sterile glass tube without breast sanitization to give a more representative analysis of the bacteria ingested by the suckling neonate. Samples were stored at −70 °C. Based on a questionnaire, sociodemographic and clinical information were recorded as well as anthropometric, and biochemistry data were collected. For each participant, a 1-week food frequency questionnaire, designed to obtain information about eating habits, was applied.

### 2.3. DNA Extraction

For DNA extraction, 200 mg of MS and IS samples were processed using the FavorPrep™ Stool DNA Isolation Mini Kit (Cat. FASTI 001-1, FAVORGEN© Biotech Corporation, Zhunan, Taiwan) following the manufacturer’s instructions. For HM samples, 1 mL was processed using the FavorPrep™ Milk Bacterial DNA Extraction Kit (Cat. FAMBD 001, FAVORGEN© Biotech Corporation, Zhunan, Taiwan). DNA integrity was assessed by 0.5% agarose electrophoresis gel and the purity was evaluated using 260/280 and 260/230 absorbance ratios, measured in the NanoDrop Lite Spectrophotometer equipment (Thermo Scientific, Waltham, MA, USA).

### 2.4. Amplification of the Bacterial V3-16S rRNA Gene Region

Microbiota composition was established by sequencing the V3 polymorphic region of the bacterial 16S rRNA gene in each sample, contained in an ~281 bp PCR product amplified using V3–341F forward primer (set of barcodes) complementary to positions 340–356, and V3–518R reverse primer complementary to positions 517–533 of the *Escherichia coli* 16S rDNA molecule *rrnB* GenBank J01859.1. The thermocycling program was 3 min at 98 °C; 25 cycles (12 s to 98 °C, 15 s to 62 °C, and 10 s to 72 °C); and 5 min at 72 °C. The PCR product was visualized in 2% agarose gel. The amount of each amplicon was estimated by densitometry, using the Image Lab v.4.1 program, and a final library was made by mixing equal amounts of amplicons [36].

### 2.5. High-Throughput DNA Sequencing

The final library was purified using 2% agarose gel stained with SYBR GOLD DNA (E-Gel™ EX, 2%, Invitrogen™, Cat. G401012, Waltham, MA, USA). The DNA library concentration and final size fragment were measured with 2100 Bioanalyzer Instrument (Agilent Technologies, Santa Clara, CA, USA), the resulting average size of the library was 256 bp. Emulsion PCR was carried out using Ion PGM HI Q view OT2 Kit (Cat. A29900, Life Technologies, Carlsbad, CA, USA), according to the manufacturer’s instructions. Sequencing was made using the Ion PGM HQ view SEQ kit (Cat. A30044, Life Technologies, Carlsbad, CA, USA) and the Ion 318 Kit V2 Chip (Cat. 4488146, Life Technologies, Carlsbad, CA, USA), and the Ion Torrent PGM system v4.0.2. After sequencing, the readings were filtered by the PGM software to remove the polyclonal (homopolymers > 6) and low-quality sequences (quality score ≤ 20).

### 2.6. Metabolite Analysis by ESI FT-ICR MS

For fecal samples, 100 mg were used while 100 µL were used for human milk samples. To prepare the sample, 1 mL of deionized water (Merck/Millipore Simplicity UV. Burlington, MA, USA) and 100 µL of 0.1M HCl (Meyer, Mexico City, México, Cat. 8110-1000) were added, and the mixture was vortexed for 15 s. Afterward, the samples were centrifuged at 13,000 rpm for 10 min. A 200 µL aliquot from each supernatant was filtered through 0.45 µm PTFE membrane (Cat. 1100213631, Navigator Lab Instrument Co., Ltd. Tianjin, China) and 50 µL injected directly into the mass spectrometry equipment. High-resolution mass spectra were acquired on a SolariX XR Fourier Transform Ion Cyclotron Resonance Mass Spectrometer (FT-ICR MS) equipped with a 7 Tesla superconducting magnet, and an Apollo II electrospray (ESI) (Bruker, Bremen, Germany). The ESI was calibrated in positive and negative mode with sodium trifluoroacetate solution. The spectra were exported to peak lists with a cut-off of signal-to-noise ratio (S/N) of 6 using the DataAnalysis 5.2 software. Bruker Compass MetaboScape 2022 b v.9.0.1 software was utilized for the identification of metabolites present in each sample.

### 2.7. Bioinformatic Analysis

Amplicon Sequence Variants (ASVs) were determined from reads that satisfied the quality criteria using the QIIME2-2023.5 pipeline [37], with the option “–p-trunc-len” set to 173 nucleotides. Representative sequences were taxonomically annotated with Greengenes2 database [38]. Further analyses were performed with R 4.3.3 [39] into RStudio 2023.12.1 + 402 IDE [40]. Data were imported into R with qiime2R 0.99.6 package [41]; phyloseq 1.46.0 package [42] was used to provide data structure and useful basic functions for the analysis of microbial communities. For intra-sample diversity the observed, Shannon, Simpson, and Fisher indexes were calculated. Analysis of the inter-sample diversity was carried out with UniFrac distance, and Non-Metric Multidimensional Scaling (NMDS) ordination with vegan 2.6-4 package [43]. Core microbiota heat map (30% prevalence, 5% detection) and Spearman’s rank correlation of bacteria with variables (anthropometric, biochemistry, and dietary data) were elaborated with microbiome 1.22.0 [44], and ComplexHeatmap 2.16.0 [45] packages. Differential abundance analysis was performed with DESeq2 1.40.2 [46]. Sourcetracker2 gibbs sampler [47] was used to check similarities among sample types.

For peak selection, Bruker MetaboScape 2022b software v.9.0.1 was used with T-Rex 2D algorithm, with 60% of minimum feature presence in sample types, and both minimum # features for extraction and presence of features in minimum # of analytes set to 41/66 analytes, signal noise threshold was set to 500,000 of intensity. Also, MetaboScape annotation feature was applied, using the Human Metabolome Data Base (HMDB) as a target list. Further analyses were carried out with RforMassSpectrometry 0.1.4 [48] package. Data were normalized with Variance Stabilizing Normalization using vsn 3.68.0 [49]. The zero values were imputed to the minimum value, and metabolites with 55% prevalence among samples were selected for hierarchical clustering heatmap with QFeatures 1.11.2. Principal Component Analysis was performed with normalized data. Additional molecular formula prediction was made and Kendric, Van Krevelen, and NOSC plots were made with ftmsRanalysis 1.1.0 [50].

The optimal number of clusters and partitioning around medoids were performed with fpc 2.2–11 [51]. Data were managed with tydiverse 2.0.0 [52]. Figures were elaborated with ggplot2 3.4.4 [53], ggpubr 0.6.0 [54] and ggdendro 0.1.23 [55]. For the significance level, *p*- and *q*-values < 0.05 were considered as statistically significant. Metabolome and microbiota data were normalized with the Centered Log Ratio (CLR) and Multiblock Sparse Partial Less Squares Differential Analysis (sPLSDA), also known as DIABLO including in miXomics 6.24.0 [56] package. Model tuning was performed, for component selection, 10 components were tried using the Mfold validation parameter. Repeated cross-validation was applied to select the number of features in the model with centroid distance to estimate the classification error rate.

## 3. Results

### 3.1. Characteristics of the Participants and Studied Samples

We studied 21 participant mothers, most of them COVID-19 vaccinated, who satisfied the selection criteria (Table 1, Appendix A). We had 20 mothers at month 0 of delivery, ten of them quit and one new mother with 4 months of delivery was included at time 4 months. Women had an average age of 34 years old, with a gestational age of 38.7 weeks. Most births were non-elective C-section, and skin-to-skin contact was prevalent among binomials. For the mothers, the anthropometry shows that most of the participants were overweight in the immediate post-partum or pregestational period. The blood test showed a significant reduction in serum triglycerides, phosphatase, and C-reactive protein, from 0 to 4 months, while albumin increased. The same increase was observed for the Na+ and K+ levels (Table 1). The results of the 1-week food frequency questionnaire applied to the participants showed a slight statistically significant increase in energy from linoleic acid consumption at month 4 compared to month 0. The same study suggests that there is no change in the overall energy intake of the participant mothers (Appendix A). In the case of infants, they were exclusively breastfed, with a weight close to 3 kg at birth and a height of approximately 50 cm. In addition, the somatometry suggests that infants are healthy. The rest of the test reflects expected changes for a healthy group of infants (Table 2, Appendix A). The number of samples collected in this study was 25 MS (15, 0 months, and 10, 4 months), 26 HM (18, 0 months, and 8, 4 months), and 32 IS (23, 0 months and 9, 4 months). The imbalance in the number of samples is due to the fact that some mothers declined to provide HM or MS samples, and also many of the participants abandoned this study for the 4 months.

### 3.2. The Bacterial Composition among Samples Suggests Vertical Transmission

The bacterial composition of the microbiota was determined in the DNA extracted from the samples using semiconductor massive DNA sequencing of V3-16S rDNA libraries. The results show close to 5 million reads, with a mean of approximately 60,000 reads per sample (Appendix A). The bacterial composition of the microbiota found in the MS, HM, and IS was assessed using alpha and beta diversities, plus relative abundance metrics. The alpha diversity shows a tendency of a higher observed number of species in the MS samples than in the HM and the IS samples. The Shannon and Simpson indices show the diversity, and the dominance are higher in MS than in the rest of the samples, while the evenness reported by the Fisher index shows that only the IS-4M is lower (Figure 1A, Appendix A). The beta diversity distance matrix that evaluates the microbial communities’ dissimilarities displays seven clusters of samples. Among them, the first one groups mainly MS-0 and MS-4M; and the second and third clusters which are more similar in the graphic are group HM-0M, and IS-0M, and IS-0M and IS-4M (Figure 1B, Appendix A). The relative mean abundance, assessing the abundance of ASVs at the phylum level, and the beta diversity show a predominance of *Firmicutes*_D for the HM-0M, IS-0M, and HM-4M, of *Firmicutes*_A for MS-0M and MS-4M. The *Actinobacteriota* phylum is remarkably abundant in IS-4M, while the Proteobacteria phylum is higher in HM-4M and IS-0M. On the other hand, *Patescibacteria* is a phylum that is almost exclusively high in HM-0M. The beta diversity dendrogram shows the bacterial communities for HM-0M and IS-0M are more similar; in a different branch, the bacterial community of HM-4M is shown as more similar to the stool communities in IS-4M, MS-0M, and MS-4M (Figure 1C).

### 3.3. HM Drives Early Infant Gut Colonization with Drastic Changes in Bacterial Genera Occurring during the First 4 Months of Life

Further exploration of the main bacterial taxa unwrapped similarities among the studied group at the genera level. Noteworthy is the shared abundance of *Streptococcus*, *Lactobacillus*, and *Bifidobacterium* between HM and IS both at 0 and 4M. Moreover, *Bifidobacterium* was the most dominant bacteria in the IS-4M group, while the GWA-2-37-10 genus of the Patescibacteria phylum spiked in HM-0M samples (Figure 2A, Appendix A). The hierarchical clustering heatmap of the core microbiota of genera with 30% prevalence among samples in at least 5% relative abundance, clearly separates by similarity the studied groups MS-0M and MS-4M from the rest, where the HMs and the IS-0M are more similar (Figure 2B). In addition, it is shown that the similarities between HM-0M and IS-0M are in the great variety of bacteria including *Paracoccus*, *Cutibacterium*, *Staphylococcus*, and *Sphingomonas*, suggesting vertical transmission of bacteria through HM and further colonization of the infants’ gut. The core microbiota also depicts a myriad of bacterial genera present in MS (for instance *Faecalibacterium*, *Bacteroides*, and *Escherichia*), which are also abundant in the IS-4M samples, indicating a closer similarity between MS-4M and IS-4M, with some genera like *Rothia*, *Veillonella*, and *Pauljensenia* being shared between HM-4M and IS-4M (Figure 2B). Additional concordant results are shown by the Sourcetracker analysis (Appendix A). Finally, we explored the bacterial genera changing across time (0M and 4M) using DESEq2 analysis and found a larger number of bacteria differentially abundant in IS samples, indicating that the infants’ gut undergoes multiple changes in the 4-month studied period (Figure 2C, Appendix A). Additional data on the sequencing depth and ASV distribution are in Appendix A.

### 3.4. Sample m/z Profile Reveals a Two-Group Clustering Corresponding to HM and MS/IS

For the metabolome workflow, a matrix containing *m*/*z* peaks per sample was used. First, the mean relative abundance of *m*/*z* peaks was calculated for the studied groups, and the 15 most abundant peaks were annotated (Figure 3A, Appendix A). In this sense, it is worth mentioning the high abundance of a 707.22917 *m*/*z* peak in HM at 0M, 4M, and MS-4M; the rest of the abundant peaks were not shared among samples (Figure 3A). Continuing with the masses exploration, Principal Component Analysis (PCA) was applied to VSN (Variance Stabilizing Normalization) transformed metabolite matrix. According to this ordination, HM samples are aligned to the left side of the graph, being HM-4M at the upper part; while both MS and IS appear on the right side, with both MS-4M and IS-4M at the upper and the 0M samples mostly at the bottom (Figure 3B). Regarding the *m*/*z* peaks, they formed two clusters, predicted by k-medoids, which also correspond to HM (left side) and MS/IS (right side) (Figure 3C, Appendix A). The latter can be confirmed as HM had a higher total intensity and a greater number of peaks, compared to the other two sample types (Appendix A).

### 3.5. Metabolite Identification Shows a Complex Mixture of Molecules Conforming the Samples

The *m/z* peaks were identified using the Human Metabolome Data Base (HMDB). This provided the molecular formula and name of the metabolites, and then, molecular classes were established according to the literature [57] in amino sugars, carbohydrates, lipids, phytochemicals, and proteins. Also, metabolites not considered in these boundaries were grouped in the “Other” category. A Kendric mass plot was then created to analyze changes in the compounds present in the EMP (Figure 4A). Patterns forming lines in the plot mean they are related to each other. It is observed that certain lipids are arranged in a diagonal line, indicating they belong to the same class of long-chain fatty acids and are consecutively elongated. Similarly, proteins exhibit the same behavior, with those located from the bottom left corner to the middle. Interestingly, metabolites that are not classified within the boundaries are also found to be integrated into the protein diagonal line (Figure 4A). Furthermore, the Nominal Oxidation State of Carbon (NOSC) was calculated to detect changes in hydrogenation and dehydrogenation reactions across the studied groups. All sample types had higher abundance molecules at NOSC values of 0, indicating the abundance of carbon-to-carbon bond compounds in the EMP. Remarkably similar sample types display more similar curve patterns at time 0 and 4 months. Also, HM and IS shared similar curve patterns (Figure 4B). In our work, it is observed that molecules with NOSC values less than zero were identified as more protonated (-H). Additionally, a Van Krevelen plot (Figure 4C) was used to show the relationship between different molecular classes and their structural modifications, which can be identified as diagonal and vertical patterns occurring in the plot. The plot identified a high abundance of molecules outside the boundaries (Other), followed by proteins and lipids which are the main classes present in the samples. Also, the patterns in the metabolites present in the samples suggest an increase in molecular complexity as indicated by changes in the oxygen-to-carbon and hydrogen-to-carbon ratios (Figure 4C). This study identified metabolites that were present in at least 55% of the samples (Figure 4D). A hierarchical clustering heatmap was used to determine the features that characterized the different groups. The HM samples had a high abundance of neocasomorphin. The IS-0M and MS-0M samples are grouped with a high abundance of motolimod, B-trisacharide, oxidized phosphatidylglycerol, and an oligopeptide. IS-4M, MS-4M are clustered together due to their content in isomaltulose, gambogic acid, and an unidentified mass metabolite (Figure 4D).

### 3.6. Multiblock sPLSDA Model Reveals the Dynamics between Bacteria and Metabolites in the Entero–Mammary Pathway

The multiblock sPLSDA model was applied for the integration of microbiota and metabolome data (Appendix A). The loadings or main variables contributing to the model components were calculated to identify the feature and its corresponding associated group; this is helpful in finding differential features for the studied groups. A total of 15 bacteria and 60 metabolites were obtained (Appendix A). For the microbiota block, three unassigned ASVs belong to the HM-0M group. Further BLAST alignment to the NCBI database revealed these unassigned ASVs corresponded to sequences in the *Homo sapiens* chromosomes 16, 21, and 22. The IS-0M group was characterized by *Staphylococcus*, *Enterococcus* (both *Firmicutes*_D phylum), *Sphingomonas*, *Bradyrhizobium* (both *Proteobacteria* phylum), *Bifidobacterium* (*Actinobacteria* phylum), *Nanosyncoccus* (*Saccharibacteria* phylum) and *Methylobacterium* (*Patescibacteria* phylum). The important bacteria characterizing the IS-4M group were *Escherichia* (*Proteobacteria*) and GWA2-37-10 (*Patescibacteria*). For the case of HM-4M, the relevant bacteria were *Acinetobacter* (*Proteobacteria*) and *Cutibacterium* (*Actinobacteria*) (Figure 5A). Next, in the metabolome block, half of the 60 metabolites were identified in HMDB, the remaining unidentified masses are shown as *m/z* values. Interestingly, every component was almost uniquely described by a sample type (Component 1, MS-4M; Component 2 IS-0M; and Component 3, HM-0M, and HM-4M) (Figure 5B). Additionally, relevant identified metabolites for IS-0M were foeniculoside V, flavonoid-7-o-glycoside (5-Methyleriodictyol 7-[glucosyl-(1→4)-galactoside]), N-acetylmannosamine, lappaol B, glucosylgalactosyl hydroxylysine and 2-arylbenzofuran ((7′R,8′R)-4,7′-Epoxy-3′-methoxy-4′,5,9,9′-lignanetetrol 9′-glucoside). The IS-4M group was defined by vignatic acid B and gambogic acid. For the HM-4M, important metabolites were sulfoxide (Propyl 1-(propylsulfinyl)propyl disulfide), citbismine D. For MS-0M, only kahweofuran was of relevance and for MS-4M, pisumoside B, stercobilingen and cholyltryptophan described the samples (Figure 5B). Further exploration of both microbiota and metabolome blocks features are detailed in Appendix A. Finally, a Canonical Correlation Analysis (CCA) was applied between microbiota and metabolome loadings, and a weighted network was generated from the results (Figure 5C). Interestingly, the three unassigned bacteria and GWA-2-37-10 correlated positively to unidentified metabolites and diarylethter ((E)-1-(4-((6-Amino-5-((methoxyimino)methyl)pyrimidin-4-yl)oxy)-2-chlorophenyl)-3-ethylurea). The other positive correlation was the bacteria *Nanosyncoccus* with bosentan, hydroxy bosentan and an unidentified metabolite (*m/z*: 406.1392). Other interesting correlations, in the negative weight, occurred between cholytryptophan and three genera (*Staphylococcus*, *Cutibacterium* and *Sphingomonas*), and the metabolite stercobilingen with *Bradyrhizobium*, *Staphylococcus*, *Cutibacterium* and *Sphingomonas* (Figure 5C).

## 4. Discussion

In this study, the bacterial composition of the microbiota and the metabolites present in mother stool (MS), human milk (HM), and infant stool (IS) of Mexican mothers and infants was investigated using semiconductor massive DNA sequencing of V3-16S rDNA libraries, and FT-ICR mass spectrometry, respectively. These analyses provided information about the importance and impact of human milk on the healthy maturation and development of the infant. With this purpose, data from bacterial communities and the host’s metabolome collected immediately after delivery and four months post-partum were incorporated, offering a more robust perspective into the roles of human milk in child maturation and development.

We found valuable insights into the bacterial composition of the microbiota in the samples. It is notable that the main phyla related to sample clustering were *Firmicutes*_A in HM and IS-0M, *Firmicutes*_D in MS, and *Actinobacteria* in IS-4M. An extensive review of HM has reported 820 species present in this fluid, highlighting that the majority belong to *Proteobacteria* and *Firmicutes* phyla [58]. Also, a previous study has reported that HM of Mexican women had lower *Firmicutes* in those with current BMI ≤ 24.9, compared to BMI > 25 [59]. In another study conducted in the Dubai (UAE) non-westernized population [60] has linked a calorie-rich diet to an increase in the members of this phylum in HM. These results are consistent with our findings on a high abundance of *Firmicutes*. Moreover, another result showed an increase in *Firmicutes* and *Actinobacteria* phyla in vitro induced by HMO (2′-fucosyllactose) in the feces of Finnish breastfed infants [61]. This could not be evaluated in our study, as our extraction method did not allow us to detect HMOs, but this might explain the increase in *Actinobacteria* in IS-4M. Additionally, previous findings in South Korean women suggest that C-section deliveries had a higher relative abundance of *Proteobacteria* and a reduction in *Firmicutes* in HM [62].

The literature also provided valuable clues to our bacterial genera exploration. First, important bacteria in IS-0M were *Lactobacillus*, *Klebsiella*, *Enterococcus*, and *Bradyrhizobium*. Interestingly, *Lactobacillus* has been extensively evaluated as a probiotic to help newborn colonization as it can alleviate inflammation symptoms. These studies were made in Germany, the USA, Australia, and Mexico [63,64]. A study conducted in the UK shows that *Klebsiella* is characteristic of preterm infants’ gut microbiota, and it may lead to infections, for instance, necrotizing enterocolitis (NEC) [65]; remarkably *Enterococcus* can prevent this affection [66]. *Methylobacterium* was differentially abundant in IS-0M according to our analysis and, interestingly, it has been reported to be part of the oral and skin microbiota [67,68], so its origin can be explained by newborn skin contact with the mother.

Following the trend found in our IS-4M; in two-year-old Canadian infants, *Bacteroides* abundance was positively correlated with cognitive and language scores; this could be due to its sphingolipid synthesis and metabolism; also, this genus seems to compete with *Streptococcus* [69] explaining the reduction in this last genus in our infant samples. *Escherichia* has been found in HM and IS samples in previous publications where mother/infant dyads were recruited at a Chinese hospital, and it can bind to immunoglobulin A [70]. Another important taxon, *Faecalibacterium*, has been reported in a study conducted in Scotland to be absent in newborns and appearing at approximately 6–7 months of age, persisting at a low level until 2–3 years of age, being essential for a healthy microbiota [71].

In the context of human milk and vertical transmission via breastfeeding, previous analyses of shared genera between HM and IS in the Dutch population [72] agree with the presence of *Pauljensenia*, but not with the presence of *Streptococcus* and *Veillonella*, which are shared in our work. Also, it has been shown that *Bifidobacterium* is bound to sIgA in HM [73], and the bacterium is transmitted by breastfeeding [74]. Furthermore, *Sphingomonas*, *Bradyrhizobium*, and other bacteria are among the core bacteriome of HM. In addition, *Staphylococcus*, *Acinetobacter*, and *Cutibacterium* were also identified as abundant bacteria in HM by DNA sequencing [22]. *Cutibacterium* (*Propionibacterium*) has also been reported to transform lactate into propionate, acetate, and CO_2_ [75], which can prevent lactate accumulation alleviating colic in infants [76]. The bacteria *Paracoccus*, *Staphylococcus*, *Bradyrhizobium*, *Acinetobacter*, found in our analysis, have also been identified by LEfSe analysis of semiconductor DNA sequencing of V3-16S rDNA libraries in Mexican HM samples [77]. In another report, *Staphylococcus* and *Acinetobacter* have been found reduced and depleted, respectively in HM of mothers from India suffering from mastitis [78]. Also, it has been suggested that *Staphylococcus* might be more abundant in the gut of multiparous mothers in the South Korean population [62].

Moreover, some bacteria are linked to women’s health, both in pregnancy and after delivery. In regard to some bacteria found in this work, *Clostridium* was associated with younger and leaner pregnant women in Guangzhou, China [79]. It is important to mention that some species of the genera *Clostridium* and *Ruminococcus* have been reclassified as *Blautia*, and the last has been suggested to play an important beneficial role as a probiotic [80]. *Holdemanella*, on the other hand, was enriched in Chinese pregnant women after probiotic supplementation with *S. thermophilus* [81]. Concerning another interesting bacterium found in this work *Agathobacter*, a previous study has found higher *Agathobacter rectalis* abundance in meconium samples when compared to mother vaginal samples and it is thought to be a protective taxon against infections [82].

In the context of SCFA, and in association with some bacteria found in this work, some reports offer intriguing perspectives. For instance, *Anaerobutyricum* and other *Firmicutes* are butyrate producers [83]. *Prevotella* is another SCFA producer and its presence in mothers’ gut has been linked with children protection against food allergies in Australia [84]. *Lachnospira* has been positively associated with polyunsaturated fatty acid intake [85]. Also, *Blautia*, *Clostridium*, *Ruminococcus*, and *Dorea* were found enriched in Gestational Diabetes in Italian women following nutritional recommendations, while *Collinsella* was reduced [86]. In another study, *Dialister* has been found to decrease in women with Gestational Diabetes in China [87].

On the metabolome topic, the high abundance of 707.22917 *m/z* peak in human milk, as well as the distinct clustering patterns observed in the PCA and k-medoids analysis, suggest unique metabolic signatures for each sample type. Further inquiry into metabolite annotation of the 707.22917, revealed the putative assignation of a compound identified as Myricanol 5-(6-galloylglucoside). According to the HMDB, this substance has been found in the cell membrane and it acts as a nutrient. A previous report indicates it can be isolated from bayberry (*Myrica* spp.) and it has been reported to reduce lipid accumulation in mice cells and zebrafish [88]. More important is the presumption of consecutively elongated long-chain fatty acids in our study, as some reports have found long-chain monomethyl branched-chain fatty acid in HM of Irish women, which might be associated with gestational weight gain [89].

Regarding the HMDB annotation of some metabolites found in this work, B-trisaccharide is a metabolite found in a variety of animal-derived foods. A previous study has found this substance in urine after galactose administration [90], and it might also be a prebiotic (HMO) present in HM [91]. Neocasomorphin, found in our HM samples, is a casein-derived peptide with opioid function [92]. Phosphatidic acids as the one identified in our analysis are a class of phospholipids and main components of the milk fat globules, also associated with proteins in the human milk aqueous phase [93]. Other metabolites were especially important at 4 months, for instance, isomaltulose is a low glycemic carbohydrate which, interestingly has been used as a sweetener agent in a variety of food products [94] and milk formula [95]. Gambogic acid is a xantanoid that has been found in *Garcinia hanburyi* plant and it has been suggested to be protective against neonatal pneumonia by avoiding LPS-induced apoptosis [96].

It is interesting to see how many metabolites of food sources were found. Kahweofuran, which characterizes MS-0M, has been tracked to coffee seeds and its presence in mothers’ intestines can be explained as they drink coffee [97]. The Propyl 1-(propylsulfinyl)propyl disulfide, has been identified in onions by LC-APCI-MS [98], and it was present in HM-4M samples in our study. The c-glycosyl 1-[(5-Amino-5-carboxypentyl)amino]-1-deoxyfructose, characterized the IS-0M samples, and it has been previously detected in milk products. This compound is a substrate by *E. coli* [99]. The terpene glycoside Foeniculoside V has been tracked to herbs and spices in the database FooDB (FOOD00866) [100]. The benzofuran Lappaol B, first isolated from the plant *Arctium lappa*, can also be found in other root vegetables [101]. The cyclic peptide Vignatic acid has been previously found in pulses [102]. The flavonoid-7-o-glycoside, 5-Methyleriodictyol 7-[glucosyl-(1→4)-galactoside], has been found in sweet oranges by UHPLC methods [103,104]. Citbismine D, an acridine-containing ketone group, has also been characterized in oranges, interestingly it is found in our HM-4M samples [105]. Urolithin a 3-glucuronide found in HM-4M, is a gut microbial-derived metabolite of ellagitannins and ellagic acid reported in raspberry with antioxidant activity [106].

Other important metabolites were differential in IS-0M and can be linked to a variety of pathways. For instance, N-Acetylmannosamine, a hexosamine monosaccharide, was found in IS-0M, and it may prevent the Salla disease/infantile sialic acid storage disease pathway in humans [107]. We also detected a glucosylgalactosyl hydroxylysine, which is produced by the catalytic activity of lysyl hydroxylase in collagen [108], explaining its presence in our samples. On the other hand, the (7′R,8′R)-4,7′-Epoxy-3′-methoxy-4′,5,9,9′-lignanetetrol 9′-glucoside, a o-glycosyl compound, has been tracked to alcoholic beverage consumption (HMDB0038710) [109]; its detection in our newborns is intriguing.

It should be considered that our metabolite annotation might not always be totally accurate as some of our relevant metabolites, i.e., Motolimod, Enkephalinamide-Leu, Ala (2)-, and 2-((5-(Diethylamino)-[1,2,4] triazolo [1,5-a]pyrimidin-7-yl) (hexyl) amino) ethanol, compared to the literature, gave findings outside of our sample context. Motolimod has been identified as a substance used for oncological medication in affections like epithelial ovarian carcinoma [110]; nevertheless, we have no report of consumption of this medication in our participant mothers. The second case is a molecule that can be generated by electrochemical oxidation and microsomal oxidation of rimonabant, a Cannabinoid 1 (CB1) used to reduce appetite [111]. The third seems to be a trapidil derivative which can affect cholesterol homeostasis [112]. These molecules, regardless of not being identified as naturally occurring in the human metabolome, have simple structures and can be derived from unknown bacterial metabolism pathways. Supporting this idea are the insights revealed by the CCA network, which positively correlated many unassigned metabolites like bosentan, hydroxy bosentan, and a diarylether with similarly atypical taxa as these three unassigned taxa were later identified as *H. sapiens*, *Nanosyncoccus*, and GWA2-37-10. Bosentan is a dual endothelin receptor antagonist, which improves oxygenation in newborns with persistent pulmonary hypertension [113], while hydroxy bosentan is derived from bosentan [114]. However, the correlation of these metabolites with *Nanosyncoccus* may suggest they are part of a common metabolic pathway in which this bacterium is participating, as may be occurring with other taxa and metabolites.

Following the bacterium–metabolite interactions, negative correlations were also important for understanding the events occurring in our mothers and infants. Pisumoside B, a diterpene glycoside has been found in pea seeds [115], and its presence has been associated with beneficial *Rhizobium* (*Proteobacteria*) in healthy plants [116]. In our study, Pisumoside B in association with other *Proteobacteria* suggests a different role of this molecule in the human body. Cholyltryptophan is a cholic acid-tryptophan conjugate, and some evidence indicates this substance is produced by *Clostridia* species. [117]. In our research, Cholyltryptophan negatively correlated with *Staphylococcus*, *Cutibacterium* and *Sphingomonas*. In the intestine, bilirubin is converted by bacteria to stercobilinogen, this compound is oxidized to stercobilin which upon reabsorption contributes to low-level inflammation in a mice model [118], Our results show a negative correlation of stercobilinogen with *Bradyrhizobium*, *Staphylococcus*, and *Shingomonas* in infant stool, and *Cutibacterium* in human milk. This suggests these bacteria may metabolize stercobilinogen to another compound.

## 5. Conclusions

In this study, we have identified similarities between the gut microbiota of infants and the composition of human milk, providing insights into the interactions during breastfeeding and the changes from birth to 4 months post-delivery. Our findings suggest the existence of a shared bacteriome in Mexican mother–infant dyads, potentially influenced by maternal lifestyle and delivery conditions. These microorganisms appear to serve as protective and probiotic agents in infants, contributing to their healthy development. The metabolite profile reflects maternal dietary habits, with some molecules serving as biomarkers of specific food consumption. The presence of these biomarkers in infants suggests their origin from breast milk. Furthermore, our investigation into bacteria–metabolite interactions elucidates microbial utilization of metabolites, revealing new potential pathways and shedding light on understudied microorganisms. Overall, our results support the vertical selective transmission of bacteria and metabolites.

## Figures and Tables

**Figure 1 nutrients-16-01990-f001:**
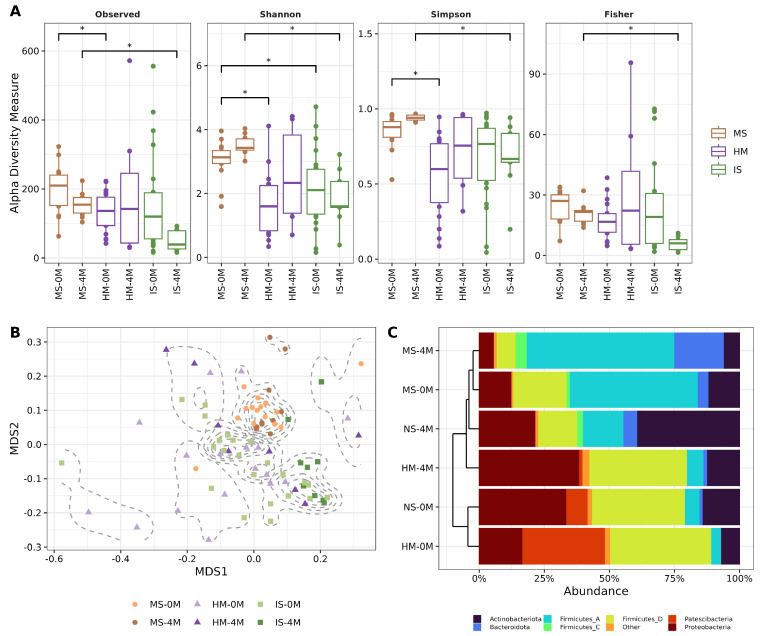
Microbiota composition and diversity in the studied samples. (**A**) Alpha diversity boxplot of the studied groups, represented by the observed number of ASVs, Shannon, Simpson, and Fisher indexes. The Y-axis indicates the alpha diversity measure and the X-axis shows the groups. Comparisons were performed by Pairwise Wilcoxon Rank Sum Test (Appendix A), and asterisks in the graph indicate *q*-values < 0.05. (**B**) Beta diversity ordination scatter plot calculated with NMDS (Non-Metric Multidimensional Scaling) from UniFrac distance. Each point in the graph represents a sample, dotted lines show the density of clusters calculated with k-medoids (Appendix A). ANOSIM (R = 0.2606, *p*-value = 0.001) was performed followed by Pairwise PERMANOVA (ADONIS) to identify differences in beta diversity. (Appendix A). (**C**) Stacked barplot of phyla relative mean abundance calculated from ASV and taxonomy tables. The left dendrogram is based on weighted UniFrac distance and indicates a hierarchical clustering among groups. The “Other” category corresponds to Phyla with less than 4% abundance.

**Figure 2 nutrients-16-01990-f002:**
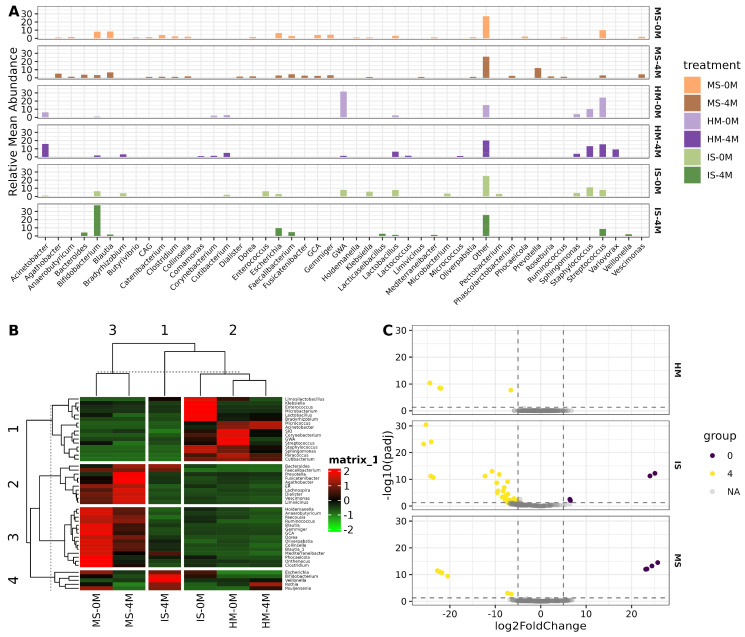
Microbial main genera conforming the samples. (**A**) Barplot of the relative abundance of genera calculated from ASV and taxonomy tables. The Y-axis shows relative mean abundance in percentage and the X-axis indicates the most abundant genera above 1% abundance. The “Other” category corresponds to genera with less than 1% relative abundance. (**B**) Core microbiota heatmap of genera with 30% prevalence among samples in at least 5% relative abundance. Column scaling was performed to facilitate the comparison among groups. (**C**) Volcano plot of the differential abundance analysis using DESeq2, showing bacteria variation in time (0 and 4 months) for each sample type. The X-axis indicates log2 fold change and the Y-axis shows −log10 of the *q*-value. (Appendix A.)

**Figure 3 nutrients-16-01990-f003:**
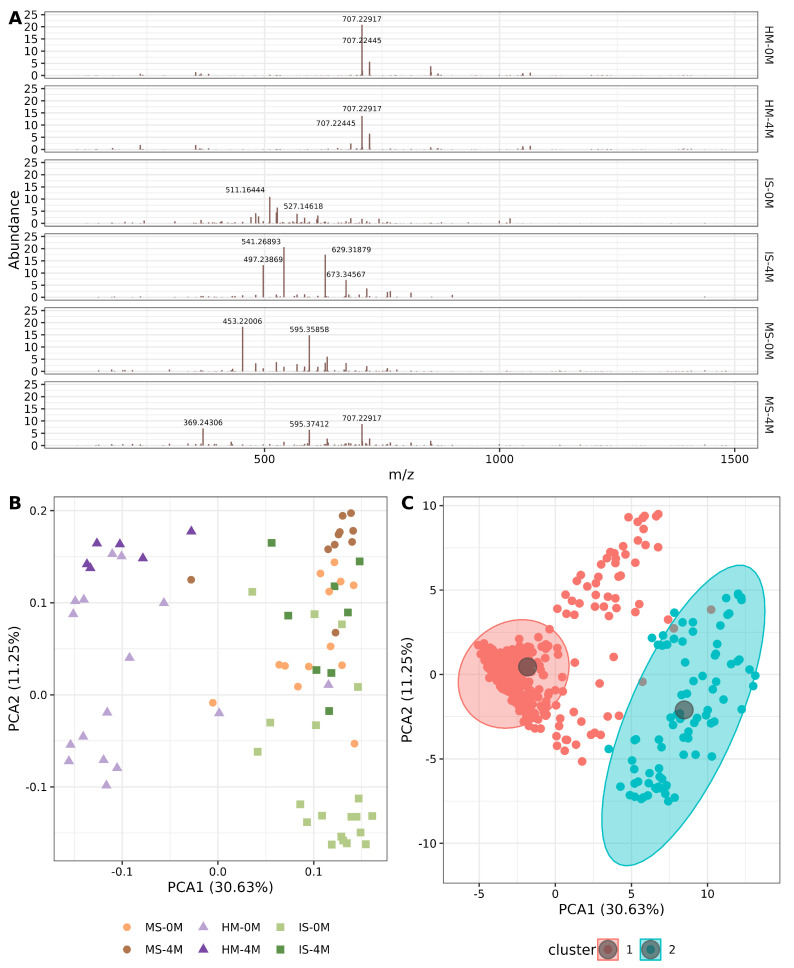
Distribution of the *m*/*z* peaks for the metabolites in the samples. (**A**) Relative abundance of *m*/*z* peaks for each group. The X-axis shows the *m*/*z* value, and the Y-axis shows the percentage of relative abundance. (**B**,**C**) show the Principal Component Analysis (PCA) biplot, explaining 41.88% of the total variance. (**B**) Loadings are represented as a scatter plot classifying the samples according to the *m/z* values, each dot represents a sample. (**C**) Scores scatter plot corresponds to the *m/z* values. Clusters are represented with color-filled areas calculated with k-medoids. (Appendix A).

**Figure 4 nutrients-16-01990-f004:**
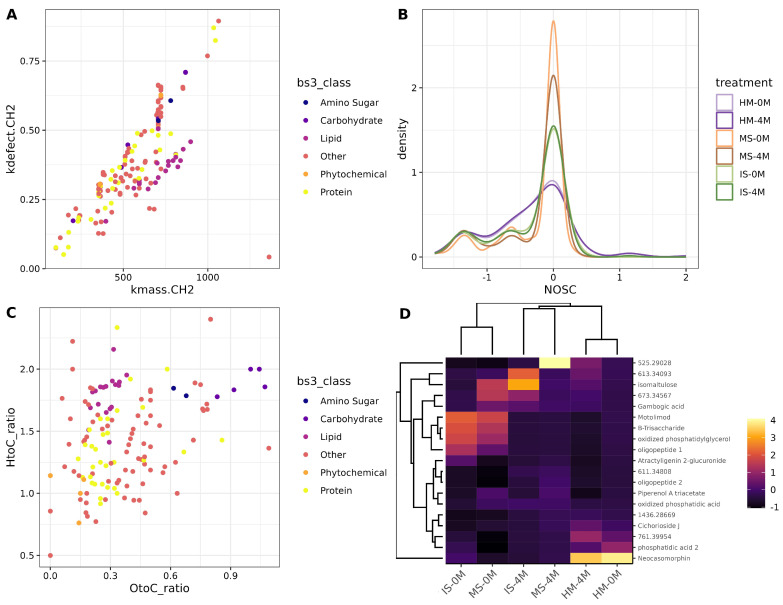
Metabolite annotation of the *m/z* values. (**A**) Kendric Mass Defect plot showing the molecular composition of identified *m/z* values. The X-axis shows the Kendrick mass versus the Y-axis indicating mass defect. (**B**) Nominal Oxidation State of Carbon (NOSC) showing the distribution of oxidation in the samples. The X-axis shows the NOSC, and the Y-axis shows the density or number of features corresponding to the NOSC. (**C**) Van Krevelen plot depicting identified *m/z* molecular classes. The X-axis shows the O to C ratio vs. the Y-axis indicating the H to C ratio (**D**). Hierarchical clustering heatmap summarizing metabolites present in 55% of the samples. Rows indicate the metabolites, while columns represent the experimental groups. Values were scaled to columns to facilitate comparisons among groups.

**Figure 5 nutrients-16-01990-f005:**
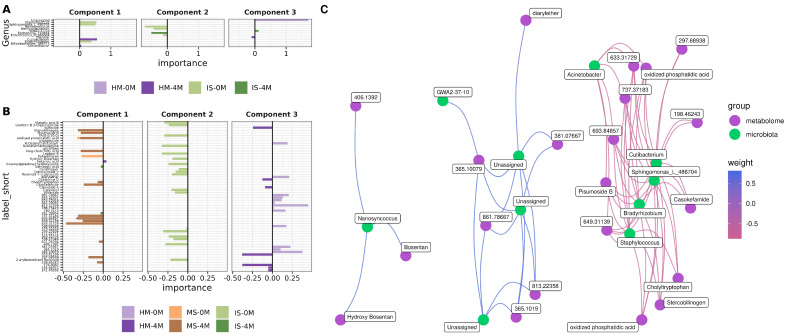
Main features (Bacteria and Metabolites) related to the entero–mammary pathway and vertical transmission by breastfeeding in the Multiblock sPLSDA model (Appendix A. (**A**) Main contributions of the microbiota block for each component. The X-axis shows the importance of the bacteria in the model; the Y-axis indicates the bacterial name. (**B**) Main contributions of the metabolome block for each component. The X-axis shows the importance of the metabolite in the model; the Y-axis indicates the metabolite name. (**C**) The correlation network between microbiota and metabolome blocks main contributions. Blue color edges represent a positive correlation while red shows a negative correlation. Canonical Correlation Analysis (CCA) was used for this purpose.

**Table 1 nutrients-16-01990-t001:** General data for mothers participating in this study.

Variable		0 Months	4 Months	*p*-Value
Number of participating mothers (*n* = 21)		20	11	---
Age (years)		31.90 (±7.49)	34.91 (±6.86)	0.291
Age range (years)		16 to 43	21 to 43	---
Average parities		2.50 (±1.15)	2.82 (±1.25)	---
COVID vaccine	Yes	18 (90.0%)	11 (100.0%)	---
	No	2 (10.0%)	0 (0.00%)	---
COVID test	Positive	1 (5.00%)	1 (9.09%)	---
Current gestation *				---
	Yes	2 (10.0%)	1 (9.09%)	
Clindamycin	No	18 (90.0%)	10 (90.91%)	---
	Gestational age (weeks)	38.60 (±1.11)	38.75 (±0.90)	0.792
	Weeks range	35.5 to 40.3	37.2 to 40.3	---
Type parities	Vaginal	1 (5.0%)	0 (0.00%)	---
	Non elective C-section	19 (95.0%)	10 (100.00%)	---
Skin-to-skin contact	Yes	13 (65.00%)	7 (63.63%)	0.951
	Average time (min)	29.77 (±17.93)	32.86 (±17.29)	0.887
	No	7 (35.00%)	4 (36.36%)	---
Anthropometry *	Height (m)	1.58 (±0.06)	1.57 (±0.07)	0.792
	Previous weight (kg)	67.02 (±14.61)	67.00 (±15.72)	0.887
	Average BMI	26.76 (±5.01)	26.88 (±5.01)	1.000
	Under weight (BMI < 18.49)	1 (5.00%)	0 (0.00%)	---
	Normal weight (BMI 18.5–24.99)	4 (20.00%)	4 (36.36%)	---
	Overweight (BMI 25.0–29.9)	10 (50.00%)	4 (36.36%)	---
	Obesity (BMI > 30.0)	5 (25.00%)	3 (27.27%)	---
	Final weight (kg)	76.24 (±13.11)	76.44 (±12.54)	0.984
Blood test *	Glucose (mg/dL)	69.90 (±13.11)	81.87 (±13.80)	**0.032**
	Cholesterol (mg/dL)	200.0 (±48.38)	199.18 (±52.04)	0.951
	Triglycerides (mg/dL)	211.30 (±79.57)	126.27 (±46.74)	**0.001**
	HDL (mg/dL)	51.10 (±14.18)	50.26 (±15.07)	0.670
	LDL (mg/dL)	106.64 (±41.71)	120.47 (±47.07) ^γ^	0.562
	Albumin (g/dL)	3.04 (±0.54)	4.53 (±0.23) ^γ^	**0.000**
	Phosphatase (U/L)	117.69 (±70.06)	86.96 (±34.27) ^β^	0.150
	Fe (µg/dL)	64.29 (±33.61) ^ζ^	59.54 (±27.99) ^α^	0.857
	HbA1c (%)	5.18 (±0.53)	4.91 (±0.40)	0.169
	CRP (mg/L)	36.99 (±21.74) ^δ^	2.91 (±2.55) ^α^	**0.000**
Electrolytes *	Na (mmol/L)	135.06 (±3.97) ^η^	143.54 (±8.64)	**0.000**
	K (mmol/L)	4.17 (±0.30)	19.07 (±47.87)	**0.001**
	Cl (mmol/L)	111.32 (±3.39)	111.15 (±5.41)	0.200
	iCa (mmol/L)	1.18 (±0.06)	1.21 (±0.08)	0.455
	iMg (mmol/L)	0.50 (±0.07) ^η^	0.56 (±0.08)	0.145
Hormone	Insulin (µU/mL)	8.43 (±5.69) ^η^	13.46 (±7.72) ^β^	**0.047**
	Osteocalcin (ng/L)	302.56 (±182.55) ^ε^	396.95 (±304.11) ^α^	0.672

*n*, sample number; min, minutes; m, meters; g, grams; mg, milligrams; µg, micrograms; mmol, millimole; mL, milliliters; µU, microUnits; ng, nanograms; dL, deciliters; L, Liters; U, Units; BMI, Body Mass Index; HDL, Hight-Density Lipoprotein; LDL, Light-Density Lipoprotein; HbA1c, Glycosylated Hemoglobin; Fe, Iron; Na, Sodium; K, Potassium; Cl, Chlorine; iCa, ionized Calcium; iMg, ionized Magnesium; CRP, C-reactive protein; percentage is shown as %; standard deviation is shown as ± values; *p*-value was calculated using the Mann–Whitney test; *p* < are considered statistically significant differences; ^α^, statistical analysis with 5 samples; ^β^, statistical analysis with 8 samples; ^γ^, statistical analysis with 9 samples; ^δ^, statistical analysis with 14 samples; ^ε^, statistical analysis with 15 samples; ^ζ^, statistical analysis with 18 samples; ^η^, statistical analysis with 19 samples. Previous weight, the data are reported by the mother before pregnancy. Final weight, post-partum weight. *, The report anthropometry, blood test, electrolytes, and hormone values at 0 months are post-partum. Figures in bold refer to values below 0.05.

**Table 2 nutrients-16-01990-t002:** General data for infants participating in this study.

Variable		0 Months	4 Months	*p*-Value
Number of participating infants (*n* = 25)		24	9	---
Gestational age (weeks)		38.47 (±1.16)	38.51 (±0.76)	0.953
Type parities	Vaginal	1 (4.16%)	0	---
	Non elective C-section	23 (95.83%)	10 (100.00%)	---
Gender	Male	10 (41.66%)	6 (66.66%)	---
	Female	14 (58.33%)	3 (33.33%)	---
Skin-to-skin contact	Yes	15 (62.50%)	6 (66.66%)	---
	Average time (min)	28.20 (±18.76) ^η^	32.50 (±19.94) ^δ^	0.742
	No	9 (37.50%)	3 (33.33%)	---
Type of feeding	BM	24 (100.00%) ^μ^	9 (100.00%)	---
Anthropometry	Weight (kg)	2.96 (±0.93) ^λ^	7.54 (±0.56) ^ε^	**0.000**
	Height (cm)	49.83 (±3.20) ^κ^	68.40 (±3.73) ^ε^	**0.000**
	Fat (%)	8.19 (±7.26) ^ζ^	91.81 (±7.26) ^γ^	0.219
	Fat-free mass (%)	13.52 (±9.60) ^ζ^	86.48 (±9.60) ^γ^	0.219
Circumferences	Head circumference	35.29 (±4.32) ^κ^	42.90 (±1.13) ^ε^	**0.000**
	Abdominal circumference	29.88 (±3.32) ^ι^	42.11 (±1.59) ^ε^	**0.000**
	Upper arm circumference	9.59 (±2.54) ^ι^	13.88 (±1.46) ^ε^	**0.000**
	Femur circumference	12.36 (±1.81) ^θ^	23.80 (±1.81) ^ε^	**0.000**
Skinfolds	Triceps skinfold	0.61 (±0.88) ^ι^	8.0 (±1.77) ^ε^	**0.000**
	Subscapular skinfold	0.55 (±0.64) ^ι^	7.75 (±4.03) ^ε^	**0.000**
Blood test	Glucose (mg/dL)	57.55 (±12.19) ^ξ^	74.67 (±5.64) ^β^	**0.012**
	Cholesterol (mg/dL)	61.70 (±17.46) ^ξ^	137.67 (±6.11) ^β^	**0.001**
	Triglycerides (mg/dL)	33.87 (±29.89) ^ξ^	148.67 (±52.79) ^β^	**0.003**
	HDL (mg/dL)	27.81 (±6.53) ^ξ^	39.96 (±10.98) ^β^	0.078
	LDL (mg/dL)	27.04 (±10.63) ^ξ^	67.97 (±4.66) ^β^	**0.001**
	Albumin (g/dL)	3.57 (±0.32) ^ξ^	4.55 (±0.18) ^β^	**0.001**
	Phosphatase (U/L)	143.71 (±59.45) ^ξ^	215.37 (±84.04) ^β^	0.134
	Fe (µg/dL)	154.20 (±34.50) ^ξ^	62.10 (±7.21) ^β^	**0.001**
	CRP (mg/L)	0.62 (±0.99) ^κ^	0.10 (±0.05)	0.125
	Po3 (mg/dL)	6 (±1.17) ^ξ^	6.3 (±1.71) ^β^	**0.395**
	BUN (mg/dL)	11.35 (±3.01) ^ξ^	7.57 (±2.95) ^β^	**0.064**
	Urea (mg/dL)	24.30 (±6.45) ^ξ^	nd	---
Electrolytes	Na (mmol/L)	130.63 (±5.31) ^ν^	143.23 (±7.05) ^β^	**0.006**
	K (mmol/L)	10.19 (±4.19) ^ν^	4.85 (±0.46) ^β^	**0.003**
	Cl (mmol/L)	111.03 (±3.30) ^ν^	112.53 (±5.58) ^β^	0.844
	iCa (mmol/L)	1.11 (±0.36) ^ν^	1.28 (±0.08) ^β^	0.497
	iMg (mmol/L)	0.58 (±0.29) ^λ^	0.61 (±0.10) ^β^	0.166
Hormone	Insulin (µU/mL)	4.68 (±2.71) ^μ^	nd	**---**
	Osteocalcin (ng/L)	402.97 (±239.98) ^θ^	238.80 (±130.01) ^ξ^	0.211

*n*, sample number; min, minutes; cm, centimeters; g, grams; mg, milligrams; µg, micrograms; mmol, millimole; mL, milliliters; µU, microUnits; ng, nanograms; dL, deciliters; L, Liters; U, Units; HDL, Hight-Density Lipoprotein; LDL, Light-Density Lipoprotein; Fe, Iron; Na, Sodium; K, Potassium; Cl, Chlorine; iCa, ionized Calcium; iMg, ionized Magnesium; CRP, C-reactive protein; percentage is shown as %; Po3, Phosphite ion; BUN, Blood Urea Nitrogen; BM, Breastmilk. Standard deviation is shown as ± values; *p*-value was calculated using the Mann–Whitney test; *p* < are considered statistically significant differences. The superscript indicates the number of samples analyzed. ^β^ (3); ^γ^ (5); ^δ^ (6); ^ε^ (8); ^ζ^ (14); ^η^ (15); ^θ^ (16); ^ι^ (17); ^κ^ (18); ^λ^ (20); ^μ^ (21); ^ν^ (22); ^ξ^ (23). Figures in bold refer to values below 0.05.

## Data Availability

Sequencing data reported in this study have been submitted to the National Center for Biotechnology Information BioProject Archive under accession no. PRJNA1086989 and the link is https://www.ncbi.nlm.nih.gov/sra/PRJNA1086989, accessed on 19 June 2024.

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
