# Peer review of "Post Natal Microbial and Metabolite Transmission: The Path from Mother to Infant"

_nutrients, 2024, doi:10.3390/nu16131990_

Round 1
Reviewer 1 Report
Comments and Suggestions for Authors
Dear Authors,
Thank you for the carefully prepared manuscript.
The study on microbial and metabolite transmission is extremely relevant and the use of the English language is appropriate. However, a minor revision considering the following points:
1. The study design is a little indistinctive. In the abstract, you mention that 21 mothers and 25 infants were included in the study. Then, in the Study design section, you mention that singleton pregnancy was an inclusion criterion. How can you have more infants than mothers if they were all singleton pregnancies? Also, in the results section, you mention that you had 20 mothers at month zero, 10 of whom completed the study and one new mother was included at 4-months. If you had 20 mothers at month 0, why did you only collect 15 samples of the mother’s stool at month and 18 samples of breast milk? Also, at 4 months you only collected 10 samples of MS, 8 samples of HM and 9 samples of IS. Why did not all participants submit their samples at all time points?
2. The latin nomenclature should be italicised throughout the text (e.g. lines 566, 568, 443, 514, 612)
3. In lines 137-138, you write that “Participants were healthy lactating women, and full-term exclusively breastfed infants” and in line 246 you write that most of them were breastfed. How do you explain this? Were all infants exclusively breastfed or not? How many of them were not breastfed? Finally, those who were not breastfed should have been used as a control group, since is makes no sense to study the shared bacteriome in breast-fed mother-neonate pairs in non-breastfed infants.
4. You concluded that the results findings indicate a shared bacteriome in breast-fed mother-infant pairs, however, I did not see that you labelled mother-infant pairs in the results. You compared the bacteriome of mother’s milk, the infant’s stool and the mother’s stool in general, so the conclusion is a little overstated.
Author Response
Review Report (Reviewer 1).
Comments and Suggestions for Authors
Dear Authors, Thank you for the carefully prepared manuscript.
The study on microbial and metabolite transmission is extremely relevant and the use of the English language is appropriate. However, a minor revision considering the following points:
Answer: We thank the reviewer for the constructive review and encouraging remarks for our work.
1. The study design is a little indistinctive. In the abstract, you mention that 21 mothers and 25 infants were included in the study. Then, in the Study design section, you mention that singleton pregnancy was an inclusion criterion. How can you have more infants than mothers if they were all singleton pregnancies?
Answer: The reviewer is right in its remark. Although we worked with singletons, as stated in the manuscript; the disbalance in the number of samples arises from the fact that the mothers of four infants declined to provide samples. However, the extra infants were included in the study.
Also, in the results section, you mention that you had 20 mothers at month zero, 10 of whom completed the study and one new mother was included at 4-months. If you had 20 mothers at month 0, why did you only collect 15 samples of the mother’s stool at month and 18 samples of breast milk?
Answer: As we mentioned before, some mothers declined to supply specific samples of stool or milk, or in some cases, the sample was not of the required quality for the study.
Also, at 4 months you only collected 10 samples of MS, 8 samples of HM and 9 samples of IS. Why did not all participants submit their samples at all time points?
Answer: Thank you for your feedback, the reasons of this disbalance in the number of samples have been explained above also, and some mothers abandoned the study. We made changes to the new version of the manuscript trying to clarify the sample size (Lines 252 to 253).
2. The latin nomenclature should be italicised throughout the text (e.g. lines 566, 568, 443, 514, 612).
Answer: We thank the reviewer for bringing into our attention this issue. We consulted the literature about the matter, and all names have been italicized in the text as requested (Lapage et al., 1992).
3. In lines 137-138, you write that “Participants were healthy lactating women, and full-term exclusively breastfed infants” and in line 246 you write that most of them were breastfed. How do you explain this? Were all infants exclusively breastfed or not? How many of them were not breastfed? Finally, those who were not breastfed should have been used as a control group, since is makes no sense to study the shared bacteriome in breast-fed mother-neonate pairs in non-breastfed infants.
Answer: The reviewer is right on this remark on the information of line 246. There was a mistake on this explanation, we have consulted with the healthcare personal, and all the children of the study were exclusively breastfed. We have made the corresponding change in the manuscript (Lines 246-247).
4. You concluded that the results findings indicate a shared bacteriome in breast-fed mother-infant pairs, however, I did not see that you labelled mother-infant pairs in the results. You compared the bacteriome of mother’s milk, the infant’s stool and the mother’s stool in general, so the conclusion is a little overstated.
Answer: Thank you for this relevant matter in our work. We were not able to analyze mother-infant binomia as dependent samples, in a paired test as the sample size was not sufficient and only allowed for independent comparisons. However, the Sourcetracker analysis (Figure S5) supports the idea of vertical transmission, as bacterial contribution from HM is high at first stage of life (~4 hours after breastfeeding) and goes down when samples were collected at 4 months. The statement in the conclusion has been softened as consequence of the reviewer remark (Lines 629-630).
References
Lapage SP, Sneath PHA, Lessel EF, et al., editors. Washington (DC): ASM Press; 1992.

Reviewer 2 Report
Comments and Suggestions for Authors
Dear Authors,
submitted manuscript is very impressive.
The significance of the research is sufficiently summarized in the introduction, which is supported by adequate citations. Also, methodology part is described at the required level with all necessary information.
Result part - Table 1: Explain how you explain the large standard deviations in the individual parameters of the tested mothers?
Table 2: the same question as is in Table 1
Line 475 - "Firmicutes" and other latin terms must be in italic - correct it in all manuscript
Author Response
Review Report (Reviewer 2).
Comments and Suggestions for Authors
Dear Authors, submitted manuscript is very impressive.
The significance of the research is sufficiently summarized in the introduction, which is supported by adequate citations. Also, methodology part is described at the required level with all necessary information.
Result part - Table 1: Explain how you explain the large standard deviations in the individual parameters of the tested mothers?
Answer: Regarding the large SD in tables 1 and 2, we think it is due to two main factors. First, we have no control of the individuals lifestyle, as well as individual genetic variations, as we work with biological samples from a Mexican free-life population. Second is, our sample size is limited, so we have no way to mitigate the natural occurring variance associated to our design study. Probably including more individuals would result in a more representative sample and thus, lower variance. However, addressing the high variance, we made sure to perform the appropriate (non-parametric) statistical tests.
Table 2: the same question as is in Table 1
Answer: Regarding the large SD in tables 1 and 2, we think it is due to two main factors. First, we have no control of the individuals lifestyle, as well as individual genetic variations, as we work with biological samples from a Mexican free-life population. Second is, our sample size is limited, so we have no way to mitigate the natural occurring variance associated to our design study. Probably including more individuals would result in a more representative sample and thus, lower variance. However, addressing the high variance, we made sure to perform the appropriate (non-parametric) statistical tests.
Line 475 - "Firmicutes" and other latin terms must be in italic - correct it in all manuscript
Answer: Thank you for bringing into our attention this issue, we consulted the literature about the matter. Now you should find the new manuscript with the correct italicization of all latin terms.
Lapage SP, Sneath PHA, Lessel EF, et al., editors. Washington (DC): ASM Press; 1992.
